# Barnase-Barstar Pair: Contemporary Application in Cancer Research and Nanotechnology

**DOI:** 10.3390/molecules26226785

**Published:** 2021-11-10

**Authors:** Olga Shilova, Polina Kotelnikova, Galina Proshkina, Elena Shramova, Sergey Deyev

**Affiliations:** 1Shemyakin & Ovchinnikov Institute of Bioorganic Chemistry, Russian Academy of Sciences, 117997 Moscow, Russia; kotelnikova@phystech.edu (P.K.); gmb@ibch.ru (G.P.); shramova.e.i@gmail.com (E.S.); 2Center of Biomedical Engineering, Sechenov University, 119991 Moscow, Russia; 3Research Centrum for Oncotheranostics, National Research Tomsk Polytechnic University, 634050 Tomsk, Russia

**Keywords:** barnase, barstar, cancer therapy, nanoparticles

## Abstract

Barnase is an extracellular ribonuclease secreted by *Bacillus amyloliquefaciens* that was originally studied as a small stable enzyme with robust folding. The identification of barnase intracellular inhibitor barstar led to the discovery of an incredibly strong protein-protein interaction. Together, barnase and barstar provide a fully genetically encoded toxin-antitoxin pair having an extremely low dissociation constant. Moreover, compared to other dimerization systems, the barnase-barstar module provides the exact one-to-one ratio of the complex components and possesses high stability of each component in a complex and high solubility in aqueous solutions without self-aggregation. The unique properties of barnase and barstar allow the application of this pair for the engineering of different variants of targeted anticancer compounds and cytotoxic supramolecular complexes. Using barnase in suicide gene therapy has also found its niche in anticancer therapy. The application of barnase and barstar in contemporary experimental cancer therapy is reflected in the review.

## 1. Introduction

Barnase is an extracellular ribonuclease (RNase) produced by *Bacillus amyloliquefaciens* as an active proenzyme, processed by the removal of the amino-terminal signal peptide and secreted into the extracellular space as a single-chain 110 amino acid protein. In this bacterial species, barstar, a specific intracellular inhibitor of barnase, is produced. Barstar tightly binds to barnase and thereby inhibits its intracellular enzymatic activity and protects host cells from the damaging effect of this RNase [1]. Barnase is a small single-chain protein that is known for its high stability and favorable biochemical features, namely the lack of disulfide bonds, post-translational modifications, divalent cations, or other non-peptide components required for its function [1]. 

The extracellular ribonuclease produced by *Bacillus amyloliquefaciens* (*Bacillus subtilis* strain H) was first reported in 1958 by Nishimura and Nomura [2]. The barnase activity was estimated in 1966 to be 2.0 × 10^6^ units/mg, and in 1967 the barnase cytoplasmic inhibitor, barstar, was purified and its interaction with barnase was shown to be resistant to urea, salt, or concentration change or heating of sulfhydryl reagents [3].

Both barnase and barstar were purified on a large scale as homogenous products from *Bacillus amyloliquefaciens* in 1972 [4,5], and before the establishment of molecular cloning, it required complex culturing conditions optimization, the use of a 1100-L custom-made fermenter and management of 2200 L of sorbent-medium mixtures. As a result of this laborious work, the barnase amino acid composition and sequence were estimated using hydrolysis and peptide analysis [6]. What is more interesting, the authors could further use barnase for affinity chromatography to purify barstar [5] and study the stoichiometry of the complex. It was shown that barnase and barstar form a one-to-one complex which dissociates only in the presence of strongly dissociating agents such as 5 M of guanidine HCl or 0.1% sodium dodecyl sulfate [7], suggesting that barnase-barstar interaction is extremely stable, but it is not based on covalent bonds formation. It was later shown that barstar covers the barnase active site, thus protecting the bacterial cell from ribonuclease that can accidentally be synthesized on free ribosomes in the cytoplasm, and the dissociation constant of the barnase-barstar complex is estimated to be 10^−14^–10^−13^ M [7,8]. In a series of works, it was estimated that the barnase-barstar pair is highly soluble in aqueous solutions, possesses high stability both in a complex and separately, and the members of the pair have no tendency to self-aggregation [9,10].

The cloning of the barnase gene was quite challenging, as in the absence of barstar the expression of barnase from a plasmid kills a cell, so the pioneer works were made with inactivated barnase, the gene of which was mutated in *Bacillus amyloliquefaciens* before any other manipulations [11,12]. The cloning of barstar and encoding of both barstar and barnase on a single plasmid enabled the production of active barnase in *Escherichia coli* [13]. From that time the barnase and its molecular twin Binase were used in numerous applications including folding studies [14,15], enzymatic activity investigation [16], protein-protein interactions investigation [17,18], positive selection of cloned inserts [19,20], providing male and female sterility in plants [21,22], crop defense system construction [23], specific cell depletion [24], virus elimination [25,26,27], and cancer cell killing [28,29,30,31]. 

Barnase is capable of nonspecific RNA cleavage and, being produced in the cytoplasm or being delivered into it, can decrease cell viability. The cleavage of mRNA is a straightforward toxic mechanism that differs from the mechanisms that are implemented in conventional chemotherapy [32], so an obtained resistance to chemotherapeutic agents is not likely to affect the effectiveness of barnase. Unlike RNase A, RNase 1, RNase 2, and angiogenin, barnase is insusceptible to the ubiquitous cytoplasmic ribonuclease inhibitor [33]. The extremely high affinity of the barnase-barstar complex can be used in the design of self-assembling complex therapeutic and theranostic agents. The N- and C-terminals of both barnase and barstar are located far from their interaction interface (Figure 1) [34,35], so both proteins can be fused with other functional modules without loss of function or affinity of barnase and barstar. These properties are successfully used in anticancer agent design, which we are discussing in our review.

## 2. Barnase as a Tool for Cancer Suicide Gene Therapy

The main goal of cancer therapy is the complete elimination of all cancer cells while leaving healthy cells unharmed. Gene therapy relies on a cell-specific expression of the chosen protein genes or regulatory RNA either in cancer cells or in effector cells that can be used in therapy. In suicide gene therapy, the genes that encode toxins or prodrug-converting enzymes are used to transduce cancer cells [36]. Ribonucleases are promising agents for use in anticancer therapy, as the production of RNases in the cytoplasm of eukaryotic cells leads to cell death via an apoptosis-mediated pathway [37]. Different delivery methods and promoters have been used to provide tumor-specific expression of ribonucleases, which is summarized in the recent work by E. Glinka [28]. To date, barnase has not been used directly in cancer gene therapy, but the successful use of barnase in specific cell ablation experiments shows a high potential of this enzyme for therapy.

One of the problems of toxin-based gene therapy is the susceptibility of virus-producing cells to the encoded toxin. To provide a substantial virus titer, viral vectors harboring a toxic gene must be grown under conditions that are not lethal to the host cell. It is usually achieved via the use of inducible and/or tissue-specific promoters that should be inactive in a packaging line and active in a target cell. This strategy can be complemented with the use of toxin-resistant packing cell lines, as was demonstrated by Li et al. [38].

The barnase gene was used in several types of constructs under the control of different promoters. It was shown to be compatible with the cell-specific tetracycline-controlled system. In this construct, barnase was linked to the P_TF_ promoter, the optimized tetracycline-dependent promoter for conditional toxic gene expression. Another plasmid PNSE-tTA encoded the PCMV-tTA tetracycline-dependent transactivator (tTA) under the control of a CMV promoter or the neuronal-specific enolase (NSE) promoter fragment. The hamster ovary fibroblasts, C6 rat glioma cells, 293 T human embryonal kidney cells, and PC12 rat adrenal pheochromocytoma cells were then co-transfected with the P_TF_-barnase, PCMVtTA, and pEGFP constructs. After tetracycline withdrawal, significant cell death was observed reaching 80% cell mortality. To evaluate the regulated expression of barnase in neuronal cells, stable Neuro2A cell clones were established. The expression of tTA under the control of the neuron-specific enolase promoter (PNSE) resulted in barnase expression and approximately 70% cell death in neuronal cells after removal of tetracycline [24]. This expression system was obtained for specific cell ablation and was used to obtain transgenic mice, but it shows that barnase can be used in tightly regulated expression systems which can be used in cancer therapy.

Another possible way to protect packing cells is the use of species-specific regulatory elements. Wang et al. [39] described a recombinant baculovirus accommodating the transcriptional regulatory sequence of the glial fibrillary acidic protein (GFAP) to drive the expression of the DT-A A (Diphtheria Toxin A) gene in glioma cells. Because the GFAP promoter is inactive in insect cells, they were able to generate recombinant baculovirus carrying the DT-A toxic gene. However, a cell-type-specific promoter such as GFAP displays relatively weak transcriptional activity compared with positive control sequences derived from viruses, e.g., the enhancer/promoter of human cytomegalovirus (CMV) immediate-early gene, and therefore limits its applications (discussed in [40]).

The production of a toxic protein during virus production can also be disabled by the introduction of an intron that cannot process in the packing cell line. In the recent study [40], two mammalian introns that are not functional in insect cells were inserted into the coding sequences of DT-A and barnase to disrupt their open-reading frames, which successfully abolished the expression of toxic proteins in insect cells. This enabled the production of baculoviral and AAV vectors harboring toxin genes at similar levels to those of baculoviral and AAV vectors carrying the green fluorescent protein (GFP) gene. Furthermore, these baculoviral or AAV vectors harboring the toxin genes were able to kill mammalian cells, demonstrating that the mammalian introns were spliced out properly and toxin proteins were expressed. Finally, tumor-specific promoters were cloned to drive the expression of the toxin genes, which resulted in tumor-specific cell killing [40,41].

Still, all of the mentioned methods are limited to specific expression systems. The unique advantage of barnase is the possibility of barstar coexpression that would inhibit barnase in a cell [13]. The affinity of barstar to barnase is high enough to prevent cell toxicity even in the case of barnase expression in human cells [42]. Barstar can be expressed in the cells that produce viral vectors to enable a high titer due to the low toxicity of barnase for producing cells. This approach was taken in the work of S. Agarwal et al. for the production of barnase-encoding retroviruses based on murine leukemia viruses (MLV) or human immunodeficiency virus (HIV) [43]. A separate vector was used for the delivery of barstar and eGFP to producing cells. The produced viruses were used for screening of mutations providing resistance of the target cells to the respective viruses. The resulting viruses were tested for cell viability reduction on several mammalian cell lines including the human prostate cancer cell line DU-145, the human cervical adenocarcinoma cell line HeLa, the human kidney carcinoma cell line A-498, and the male hamster lung fibroblast cell line V79-4. The extent of cell death at various virus doses varied between each diverse cell type, with each vector and between each of the vectors, but both tested constructs are effective at cell killing in all of the cell types examined [43]. The principle of barnase-encoding viral vectors production in barstar-producing cells is illustrated in Figure 2.

Both barnase and barstar are small stable proteins that were shown to be well expressed in mammalian cells, so we can expect their successful application in new gene therapy systems designs. To date, this potential is still to be realized.

## 3. The Use of Barnase Ribonuclease Activity in Cancer Therapy

The ribonuclease activity of barnase can be used for the construction of toxic proteins for cancer therapy.

However, bacterial expression of active barnase is lethal to the host cell. To overcome this problem, coexpression with barstar is used. Today, three mainline approaches to obtain active barnase are used: (1) cytoplasmic expression of barnase in the complex with barstar. In this case, a denaturation step to remove barstar from the complex, followed by renaturation of barnase, is necessary [9]. In the case of His-tagged proteins, the denaturation and renaturation steps can be performed directly on Ni-column as described in [9,29]. (2) The export of barnase to the periplasm and subsequently to the culture medium with signal sequences PelB [44], ompA [9], or PhoA [12,44,45,46]; the latter has shown the highest efficacy. (3) The enhanced export of barnase to culture medium after periplasmic expression, for example, as a fuse with maltose-binding protein accompanied by coexpression of a modified Cloacin DF13 bacteriocin [47]. In the last two methods the RNase activity of barnase molecules remaining inside the cell is suppressed by barstar.

To date, a variety of protein toxins have been used in new antitumor agents design [48]. Barnase was first applied to mammal cell lines in the form of a chimeric toxin including ribonuclease and ADP-ribosylating *Pseudomonas aeruginosa* toxin A (PE) [49]. The resulting hybrid toxin decreased the viability of PE-resistant L929 cells due to ribonuclease activity that was successfully delivered to the cytoplasm due to the PE translocation domain. Soon, another PE-based chimeric toxin was constructed, containing barnase instead of the enzymatically active domain III of PE. This protein was toxic for murine cells lines including PE-resistant hybridoma and was used for investigation of PE-induced cell translocation [50,51]. In the same work by Prior et al., it was proposed to use barnase as a toxic module for targeted cancer therapy [51]. The idea of using targeted barnase as a drug is supported by the fact that barnase can be successfully produced in bacteria in the presence of barstar. Moreover, barnase is a good component of recombinant protein, as it preserves its own function, does not interfere with the targeting of module binding activity, and can even serve as a molecular chaperone, increasing the protein solubility [52].

Barnase ribonuclease activity was successfully used to kill B-cells. The fused protein consisting of barnase and c-Myc epitope was used for targeting c-Myc-specific hybridoma. Barnase-c-myc specifically depleted anti-c-myc B cells and was not suppressed by ribonuclease inhibitor [53]. In another work, the use of barstar enabled a coupling of tumor targeting with targeting module multimerization which resulted in increased accumulated affinity (avidity), blood circulation time, and tumor accumulation [9]. The obtained HER2-specific fuse of single-chain antibody and two serially fused barnase molecules, 4D5-dibarnase, was shown to specifically interact with the surface of HER2-positive ovarian cancer cells, successfully enter a cell through internalization, and cause RNA degradation and apoptosis [29] (Figure 3). The effectiveness of 4D5-dibarnase against cancer cells was further proven in vivo in a mouse xenograft model based on an SK-BR-3 adenocarcinoma cell line [54].

In a recent work, barnase targeting to the tumor was achieved due to the encapsulation of barnase into tumor-specific liposomes. As a targeting module, a scaffold HER2-specific protein DARPin chemically conjugated to the surface of barnase-containing liposomes was used [55]. The liposomes were used in combination with the hybrid protein based on the EpCAM-recognizing DARPin EC1 and the low immunogenic variant of PE (EC1-LoPE) [56] (Figure 4). DARPin proteins were used in this work instead of antibodies because of their high affinity to target, ease of production in bacteria, and high chemical stability providing sufficient conjugation preserving their bioactivity [57]. This combination of dual targeting and complementary cytotoxic mechanisms provided the synergistic effect affecting not only the primary tumor site but also the distant metastases. 

The main task of contemporary oncology is to provide significantly improved treatment efficacy with reduced side effects through the specific delivery of drugs to targeted tissues. One of the approaches to reduce adverse reactions caused by a cytotoxic agent is based on pretargeting—the targeting concept which includes separation of the targeting moiety from the cytotoxic component and a subsequent joining of the two agents in vivo. This strategy requires the use of heterodimerization modules. Currently, four major approaches have been developed based on: (1) the noncovalent interaction between streptavidin and biotin, (2) the ability of bispecific antibodies (bsAbs) to bind both an antigen and a radiolabeled hapten, (3) the bioorthogonal inverse electron demand Diels-Alder click reaction, and (4) the hybridization of complementary oligonucleotides [58,59,60]. 

Today each of the strategies stands at a different stage of its scientific development. The barnase-barstar interface provides a new strategy of pretargeting. The principal ability to deliver a cytotoxic agent to cancer cells is shown by the example of recombinant miniantibodies specific to HER2 and fused with barstar, and a visualizing component—the recombinant fluorescent protein EGFP fused with barnase [61]. Recently, for in vivo pretargeting and successful anti-HER2 therapy, HER2-specific DARPin-barnase and barstar-coated liposomes loaded with PE40 were proposed [62,63]. The proposed method included the pre-labeling of a target tumor with hybrid protein DARPin-barnase prior to the administration of a cytotoxic component—PE40-loaded liposomes—that had barstar covalently attached to their surface (Barstar-Lip(PE40)) (Figure 5). At the first step, the pre-labeling component, DARPin-barnase, was accumulated in the tumor and cleared from the blood, and then, the cytotoxic component, Barstar-Lip(PE40), was injected. Due to the high affinity of barnase and barstar, quick and precise ligation of the two modules occurs in vivo leading to targeted cancer cells elimination. It was shown that the proposed system possesses HER2-specific antitumor activity in HER2-positive tumor-bearing mice leading to the destruction of primary tumors and distant metastases [62,63].

Summarizing the provided data, we can state that the barnase-barstar pair has potential therapeutic utility.

## 4. The Use of Barnase-Barstar Interaction for Supramolecular Complexes Assembly

Besides the regulation of ribonuclease activity, the strong interaction of barnase and barstar can be used for the assembly of supramolecular complexes consisting of proteins, and nano- and microparticles. The extremely high dissociation constant of barnase-barstar complex (10^−14^–10^−13^ M) [7,8] provides the formation of stable complexes that remain associated under severe protein denaturing conditions including high temperature, low pH as well as high salt and chaotropic agent (urea and guanidinium hydrochloride) concentrations [64]. In the same study, the stability of barnase-barstar complex was compared to the commonly used assembly systems, namely, streptavidin-biotin, antibody-antigen, and protein A-immunoglobulin, and barnase-barstar was proven to be comparable with these systems or to surpass them due to the combination of its high resistance to severe chemical perturbation and unique advantages offered by genetic engineering of this entirely protein-based system. 

As was mentioned above, the use of barnase and barstar interaction enables the assembly of multimeric targeted proteins. The use of barnase-barstar modules fused to 4D5scFv made it possible to assemble di- and trimeric complexes with increased avidity and molecular weight (81 kDa and 132 kDa versus 30 kDa of 4D5scFv monomer) [9]. Barnase and barstar were used as heterodimerising modules in a novel anticancer DNA vaccine [65,66], bringing together two units composed of an antigen presenting a cell targeting motif and an immunogenic motif. In combination with the easy cloning strategy, the heterodimeric barnase-barstar vaccine molecule provided a flexible platform for development of novel DNA vaccines [66].

The same strategy can be used to couple targeting and effector modules for cancer therapy. 4D5-dibarnase served as a targeting moiety to deliver heat shock protein Hsp70 to cancer cells through fused barstar, the resulting bifunctional complex efficiently binding to HER2-overexpressing cells and recruiting NK cells to them [67]. In recent work barnase and barstar provided the tumor targeting and regulation of CAR-T (chimeric antigen receptors) cells in anticancer therapy [68]. The designed chimeric receptor containing barstar could bind to tumor-targeted barnase instead of a tumor antigen itself. The intermediate molecule enabled the precise control of CAR-T cells activity towards cancer cells and provided the basis of a universal CAR-T cells system design.

The benefits of barnase-barstar self-assembly are most obvious for recombinant proteins, as these modules are fully genetically encoded. However, to date, the most diverse and sophisticated methods involving these modules have been developed in nanoparticles design. Convenient methods for nanoparticles modification based on physical adsorption and chemical conjugation have several disadvantages including the uncontrollable orientation and steric availability of biomolecules, the loss of components activity, and poor reproducibility. The use of complementary proteins allows for sufficient and controllable supramolecular complexes assembly. The process of assembly does not require complex equipment and reagents, so it can be useful both for diagnostic and clinical applications. The functional modules can be mixed in a test tube or perform self-assembly within an organism after consecutive injections.

Barnase and barstar can be used as “a molecular glue” in supramolecular complexes assembly. Usually, one of the named proteins is linked to a nanoparticle’s surface with the help of chemical conjugation, and its partner is genetically fused to a functional protein, for example, a single-chain antibody. The final step of nanoparticle modification is usually done via simple mixing of components in a test tube. The barnase-barstar linker provides the addition of a single molecule of a functional protein to each linker molecule on the surface of a nanoparticle (Figure 6).

This strategy was used for various types of nanoagents. The luminescent upconversion nanoparticles were designed for breast cancer diagnostics. The particles were stabilized with amphiphilic polymer and modified with barstar, which provided the attachment of a 4D5 single-chain antibody through genetically fused barnase. The resulting UCNP-Barsar:Barnase-scFv4D5 complexes were tested in an optical phantom consisting of HER2-positive SK-BR-3 cells covered by mammary gland tissue. The modified UCNP-exposed SK-BR-3 cells could be detected at a depth of 4 mm with the optical contrast as high as 10:1 benchmarked against a negative control cell line [71].

Quantum dots were also successfully modified using this system. Nanoparticles were chemically linked to either barstar or barnase and tumor targeting was achieved due to fuse proteins of a coupling partner with scFv specific for HER1 or HER2 [69]. The resulting self-assembling nanoparticles specifically bound to cancer cells expressing respective surface markers in vitro and HER2-specific nanoparticles accumulated in xenograft tumors significantly better than the untargeted nanoparticles [74].

In addition to coupling therapeutic proteins to nanoparticles, barnase and barstar can facilitate the assembly of various objects referring to the nano- and microscale themselves. This was demonstrated for fluorescent nanoparticles bound to the surface of magnetic nanoparticles. The interaction of barnase and barstar was strong enough to provide a sustainable complex formation resistant to extreme conditions including heating, low pH, and high concentrations of chaotropic agents [64]. This interface was accompanied with a HER2-targeting 4D5 module through the same barnase-barstar interaction resulting in trifunctional nanostructures including targeting a single-chain antibody, magnetic nanoparticle, and fluorescent nanoparticles. All modules retained their features in the resulting construct, providing efficient magnetic cell sorting and visualization [75].

A similar modification method was used for luminescent nanodiamonds (LND). These particles were chemically conjugated with barstar, which provided LND colloid stability in aqueous solutions. These particles were further coupled with either barnase-GFP fuse protein or gold nanoparticles linked to barnase [70]. In all cases the barnase-barstar “molecular glue” technique provided stable complexes retaining their functional features that could be obtained by simply mixing the components in a test tube.

The culmination of barnase-barstar nanoparticles modification technology development is the use of solid phase-binding peptides that help to avoid chemical conjugation at all. Either barnase or barstar can be genetically fused to a surface-binding peptide, which will help to modify a nanoparticle without chemical conjugation, and the partner protein will provide a coupling of targeting or other functional components to the particle. This approach allows one both to achieve the desired orientation of the binding module and to assemble the targeting modules according to the principle of a construction kit. It was first demonstrated for silicon nanoparticles, modified with barstar fused to SiO_2_-binding peptide (SBP-barstar). As HER2-targeting modules, the fuses of barnase with single-chain 4D5 antibody or scaffold protein DARPin-9.29 were used. These proteins provided the assembly of the outer layer of nanoparticles in a solution without using conjugation or on the surface of cancer cells (the pretargeting strategy). In the last case, the targeted proteins were first delivered to the cells and then the nanoparticles were added subsequently [73].

By replacing SBP with magnetite-binding peptide C-Mms6, the system was tailored to magnetic nanoparticles modification. C-Mms6 is a fragment of the protein which is used by magnetotactic bacteria for magnetosomes building. Magnetotactic bacteria can feel the Earth’s magnetic field due to special membrane organelles containing magnetic nanoparticles. The biomineralization of these particles is controlled by a complex of special proteins including Mms6 protein. The C-terminal peptide of this protein was genetically fused to barstar, and the resulting barstar-C-Mms6 could bind magnetite nanoparticles and stabilize them [72]. The use of the HER2-specific DARPin-barnase targeting component provided efficient self-assembly of the full construct on the surface of HER2-expressing cancer cells [72,76]. The use of this method facilitates the fast and simple acquisition of new targeted agents for targeted delivery of drugs or markers for magnetic or fluorescent cell sorting.

## 5. Conclusions

Barnase and barstar are notable both for biological activity and their physicochemical features. Their tight and strong interaction provides experimental biology with one more bio-compatible type of interaction in addition to antigen-antibody, antibody Fc-protein A/G, and streptavidin/avidin-biotin. The barnase-barstar pair represents a versatile bioconjugation platform for the design, production, and characterization of various supramolecular complexes. High-affinity interaction of barnase-barstar has already been used in the construction of innovative cancer therapy for multivalency introduction, binding of variable targeting and effector modules, and nanoparticles functionalization providing proper orientation of the targeting module. The use of self-assembling modules provides an opportunity to easily change the specificity or the effector mechanism of a complex, which provides a flexible platform for new anticancer agents design. Using barnase-barstar in pretargeting drug delivery also has great potential: first, in contrast to existing pretargeting systems based on noncovalent interaction, barnase and barstar have no endogenous inhibitors or nonspecific targets in mammals, and second, this approach is highly modular since barnase or barstar can be easily fused at the gene level with any artificial scaffold recognizing any antigens, making this approach a versatile pretargeting platform.

The ribonuclease activity of barnase is a promising source of anti-tumor agents. Cleavage of messenger RNA is a universal mechanism of cell killing, as any human cell depends on protein synthesis. The experimental data demonstrate that barnase in the form of a targeted recombinant protein that binds to the surface HER2 receptor enters the cell via receptor-mediated endocytosis and can induce apoptosis in cancer cells. Due to ribonuclease activity and the protection of virus-producing cells by barstar, barnase can also become a component of suicide gene therapy, but this potential is yet to be utilized.

One of the challenges associated with barnase-barstar-based anticancer therapy is the possible immunogenicity of these modules. Since barnase and barstar are bacterial proteins, the question about their immunogenicity arises. For some bacterial RNases (binase, RNase Sa) low immunogenicity has been shown [77,78,79], on the other hand, in a series of works, barnase and barstar are used in DNA vaccines aimed at obtaining an immune response [65,66]. Thus, the detailed study of the immunogenicity of barnase-barstar in anticancer therapy should be the subject of further research.

In conclusion, we can state that due to the unique features of the baranse-barstar pair, this complex has found its own niche in the field of cancer research and biotechnological methodology.

## Figures and Tables

**Figure 1 molecules-26-06785-f001:**
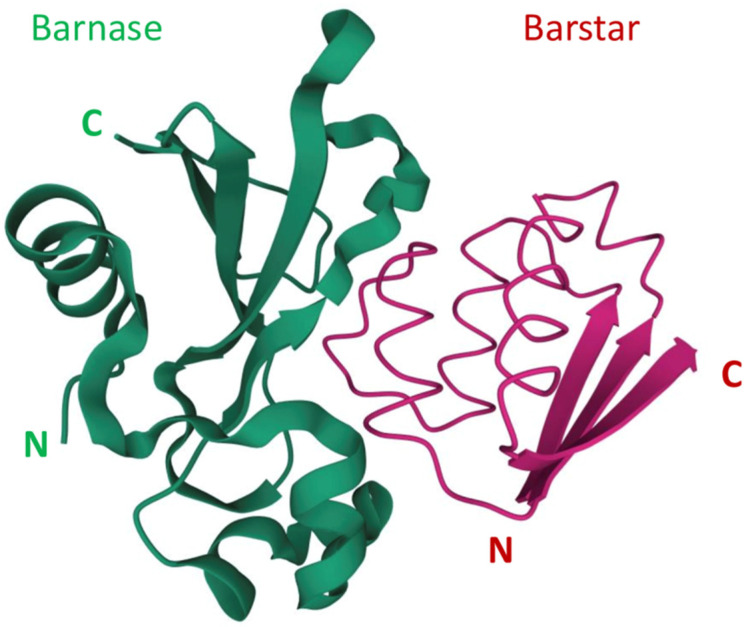
The barnase and barstar complex structure [34,35]. N and C refer to N-terminus and C-terminus respectively.

**Figure 2 molecules-26-06785-f002:**
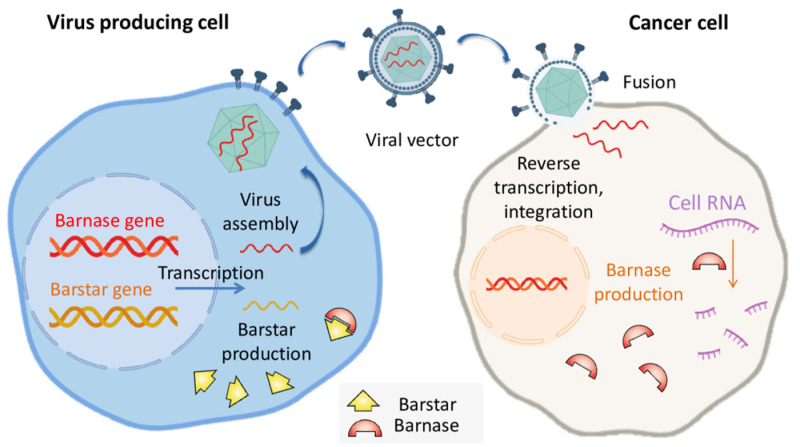
The barstar expression can protect virus-producing cells from occasional barnase expression providing a high titer of oncolytic viruses.

**Figure 3 molecules-26-06785-f003:**
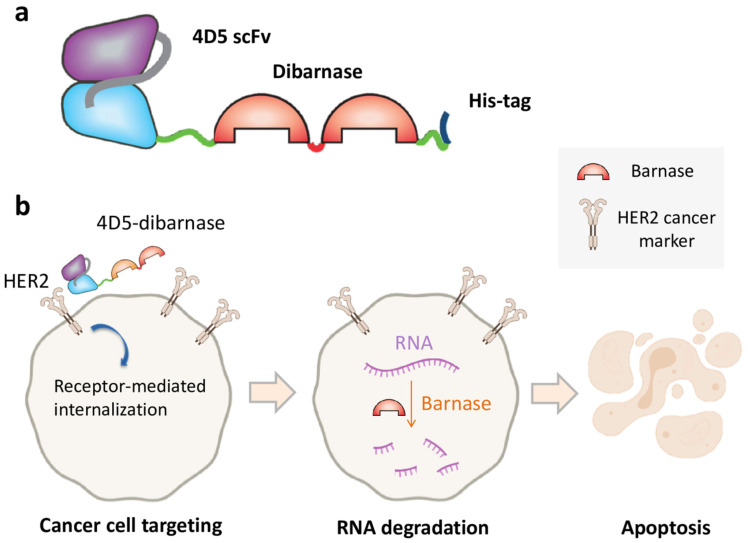
Cancer cell killing by the targeted ribonuclease. (**a**) Diagram showing the composition of 4D5-dibarnase. In this protein, HER2-specific single-chain antibody is fused to two serially fused barnase molecules provided with hexahistidine tag for protein purification. (**b**) The principle of cancer cell targeting by 4D5-dibarnase.

**Figure 4 molecules-26-06785-f004:**
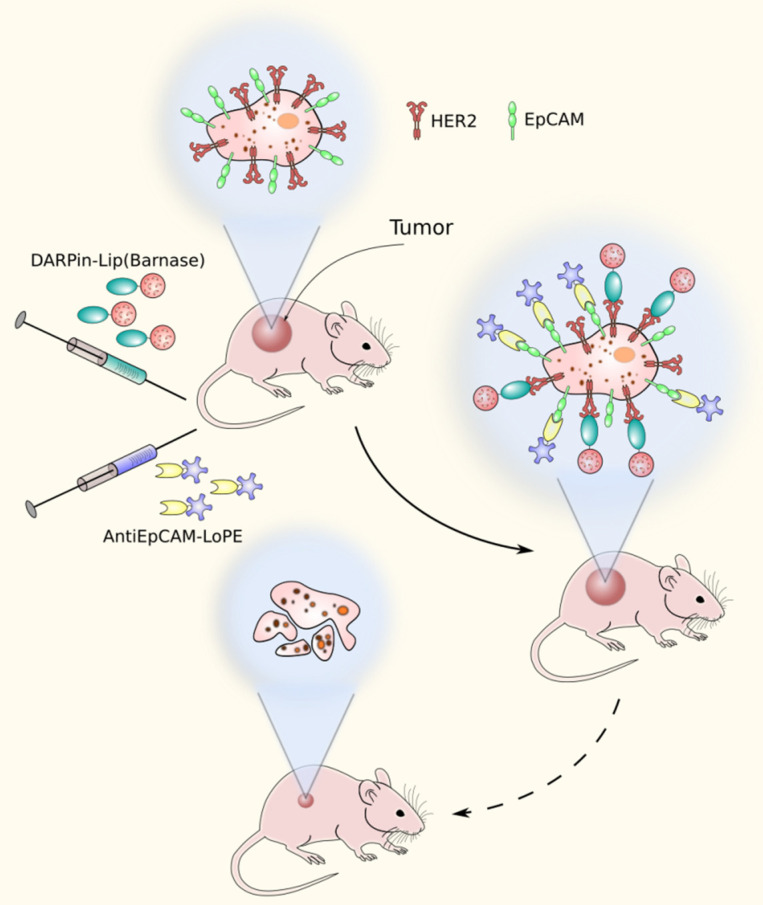
A dual-targeting strategy method based on HER2-specific liposomes containing barnase and EpCAM-specific pseudomonas exotoxin fragment.

**Figure 5 molecules-26-06785-f005:**
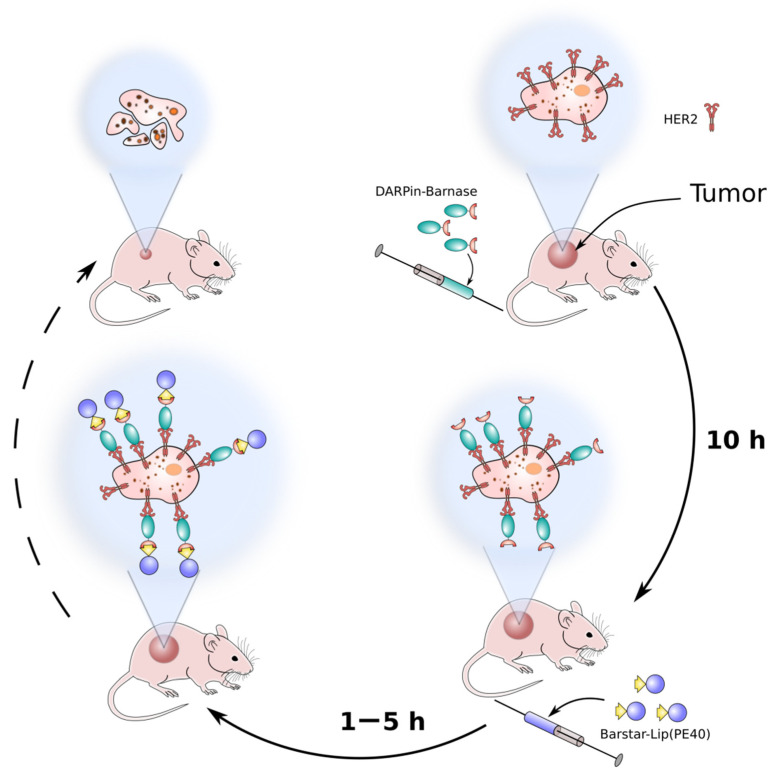
A schematic representation of the pretargeting method based on barnase-barstar pair. At the first, pretargeting step, the HER2-specific module DARPin-barnase is administrated into the tumor-bearing animals, and then, after the accumulation of DARPin-barnase in the tumor and clearance from blood, the second, cytotoxic component is injected.

**Figure 6 molecules-26-06785-f006:**
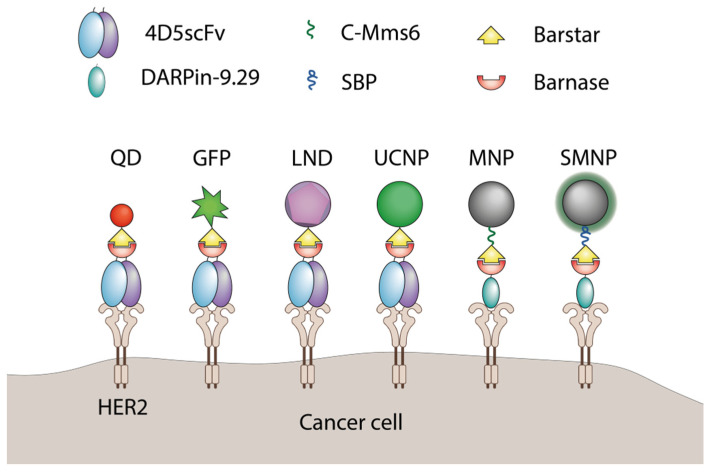
The application of barnase and barstar interaction in supramolecular complexes assembly. QD—quantum dots [69], GFP—green fluorescent protein [61], LND—luminescent nanodiamond [70], UCNP—upconversion nanoparticle [71], MNP—magnetic nanoparticle [72], SMNP—SiO_2_-coated nanoparticles [73], C-Mms6—magnetite-binding peptide [72], SBP—SiO_2_-binding peptide [73].

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
