# Peer review of "Barnase-Barstar Pair: Contemporary Application in Cancer Research and Nanotechnology"

_molecules, 2021, doi:10.3390/molecules26226785_

Round 1

Reviewer 1 Report

In this manuscript, the authors summarized the interaction of barnase and barstar, and their anticancer therapy applications. The manuscript could be published after suitable revision. Specific comments,

  • The challenges associated with barnase-barstar based anticancer therapy should be discussed in the conclusion.
  • The Recombinant expression of Barnase should be briefly discussed.
  • Recent publications such as Molecular Therapy: Methods & Clinical Development Vol. 17,  pp.378-392, June 2020, and FEBS OPEN BIO Vol.  11 , pp.93-93, Jul 2021, etc. should be included.

Author Response

We are grateful to the Reviewer #1 for a careful evaluation of our manuscript and valuable comments. We revised the manuscript accordingly. Changes in the manuscript are made in the track-changes mode.

  • The challenges associated with barnase-barstar based anticancer therapy should be discussed in the conclusion.

The challenges based on the possible immunogenicity of barnase and barstar are discussed in the conclusion (p. 12).

  • The Recombinant expression of Barnase should be briefly discussed.

The recombinant expression and purification of barnase and barnase-containing fuse proteins is discussed in the beginning of part 3 (p. 5).

  • Recent publications such as Molecular Therapy: Methods & Clinical Development Vol. 17, 378-392, June 2020, and FEBS OPEN BIO Vol.  11 , pp.93-93, Jul 2021, etc. should be included.

We are grateful to the Reviewer for the notable examples of barnase and barstar applications in cancer therapy that has not been mentioned in the paper before. We have now included them in the part 4 (p. 9).

Reviewer 2 Report

Dear authors,

my opinion about the manuscript is positive, I believe that it is suitable for publication. You can find attached my report. I only inserted some minor reviews about some sentences and references that you can easily correct. 

Author Response

We are grateful to the Reviewer #2 for the appreciation of our manuscript. We agree with all comments and revised the manuscript accordingly. Our responses are listed below.

Row 179: ‘to kill B-cells cells’. I think that ‘to kills B-cells’

We are thankful for the detailed proofreading of the manuscript. All the issues are addressed in the respective parts of the revised manuscript except the references for which the doi identifications are not available.

Row 259: ‘increased avidity’ I think that avidity is not the right word.

We used this word, because it refers to the accumulated strength of multiple affinities of individual non-covalent binding interactions. In case of multivalent complexes it seems reasonable to use “avidiy” instead of “affinity”. We have explained the word in the text.

Row 262: ‘cancer sells’ I think it is cancer cells.

Row 388: reference 8 format seems not consistent

Row 392: reference 10 lack DOI. There is also a [6] that probably is a mistake.

Row 418: reference 23 lack DOI

Row 422: reference 25 lack DOI

Row 424: reference 26 lack DOI

Row 426: reference 27 lack DOI

Row 428: reference 28 lack DOI, the year should be in bold

Row 450: reference 38 lack DOI

Row 456: reference 41 lack DOI

Row 462: reference 44 lack DOI and the format is not consistent

Row 497: reference 58 lack DOI and the format is not consistent

Row 502: reference 60 lack DOI

All the issues are addressed in the respective parts of the revised manuscript except the references for which the doi identifications are not available.

Round 2

Reviewer 1 Report

The revised manuscript could be published as it.